# FGL2/FcγRIIB Signalling Mediates Arterial Shear Stress-Mediated Endothelial Cell Apoptosis: Implications for Coronary Artery Bypass Vein Graft Pathogenesis

**DOI:** 10.3390/ijms25147638

**Published:** 2024-07-11

**Authors:** Molly L. Jackson, Andrew R. Bond, Raimondo Ascione, Jason L. Johnson, Sarah J. George

**Affiliations:** Translational Health Sciences, Bristol Medical School, Faculty of Health and Life Sciences, University of Bristol, Bristol BS2 8HW, UK; mj17303@bristol.ac.uk (M.L.J.); andrew.bond@bristol.ac.uk (A.R.B.); r.ascione@bristol.ac.uk (R.A.); jason.l.johnson@bristol.ac.uk (J.L.J.)

**Keywords:** endothelial cell, shear stress, fibroleukin, vein graft, apoptosis, NFκB

## Abstract

The sudden exposure of venous endothelial cells (vECs) to arterial fluid shear stress (FSS) is thought to be a major contributor to coronary artery bypass vein graft failure (VGF). However, the effects of arterial FSS on the vEC secretome are poorly characterised. We propose that analysis of the vEC secretome may reveal potential therapeutic approaches to suppress VGF. Human umbilical vein endothelial cells (HUVECs) pre-conditioned to venous FSS (18 h; 1.5 dynes/cm^2^) were exposed to venous or arterial FSS (15 dynes/cm^2^) for 24 h. Tandem Mass Tagging proteomic analysis of the vEC secretome identified significantly increased fibroleukin (FGL2) in conditioned media from HUVECs exposed to arterial FSS. This increase was validated by Western blotting. Application of the NFκB inhibitor BAY 11-7085 (1 µM) following pre-conditioning reduced FGL2 release from vECs exposed to arterial FSS. Exposure of vECs to arterial FSS increased apoptosis, measured by active cleaved caspase-3 (CC3) immunocytochemistry, which was likewise elevated in HUVECs treated with recombinant FGL2 (20 ng/mL) for 24 h under static conditions. To determine the mechanism of FGL2-induced apoptosis, HUVECs were pre-treated with a blocking antibody to FcγRIIB, a receptor FGL2 is proposed to interact with, which reduced CC3 levels. In conclusion, our findings indicate that the exposure of vECs to arterial FSS results in increased release of FGL2 via NFκB signalling, which promotes endothelial apoptosis via FcγRIIB signalling. Therefore, the inhibition of FGL2/FcγRIIB signalling may provide a novel approach to reduce arterial FSS-induced vEC apoptosis in vein grafts and suppress VGF.

## 1. Introduction

Coronary artery disease (CAD) is a leading cause of mortality worldwide [1], with the development of severe atherosclerotic plaques and subsequent stenosis in the coronary arteries manifesting in the form of angina, myocardial infarction and increased risk of sudden cardiac death [2]. Due to increasing socioeconomic deprivation, increased lifespan and poor lifestyle-related risk factors, CAD is an increasing burden in both under-developed and developed countries [3]. Multiple risk factors contribute to the development of CAD, including hypercholesterolemia, age, gender, smoking, diabetes mellitus, obesity and hypertension [4]. Improving lifestyle-associated risk factors such as diet modification, smoking cessation and blood glucose control significantly reduces the burden of CAD-related events [5,6]. However, the modification of lifestyle-associated risk factors is not well-implemented in the countries most affected by CAD, while the challenge of non-modifiable risk factors associated with severe CAD such as genetic predisposition, hypercholesterolaemia and ageing still remains [7]. Coronary artery bypass graft surgery (CABG) is the gold standard treatment for patients afflicted by severe/diffuse triple-vessel CAD and suffering with diabetes and low left ventricular ejection fraction [8,9]. During CABG surgery, the long saphenous vein (LSV) is the most commonly used conduit to revascularize the ischaemic heart (approx. 75% of all bypass grafts are made from LSV) due to the ease of harvest and availability [10]. Arterial grafts, e.g., the internal mammary artery or radial artery, demonstrate a superior patency rate compared to vein grafts, representing the graft of choice for the grafting of the left anterior descending (LAD) artery or neighbouring grafts, but are restricted by their limited availability and length [11,12]. However, the LSV demonstrates reduced long-term patency rates compared to arterial grafts due to the development of vascular inflammation and intimal hyperplasia, which triggers subsequent superimposed atherosclerosis and, ultimately, vein graft failure/occlusion (VGF) [13,14].

Endothelial cells (ECs) form the innermost lining of blood vessels and are essential for vascular homeostasis [15]. The quiescent endothelium acts to maintain an anti-coagulant, anti-inflammatory environment and regulates vascular tone and permeability [16]. ECs are constantly exposed to a range of mechanical forces as a consequence of their exposure to blood flow due to their position in forming the intimal layer lining the blood vessel. Fluid shear stress (FSS) is the tangential vector of the force exerted by blood flow and acts as the frictional force of blood flow per unit area parallel to the vessel wall [17]. FSS is an important regulator of vascular development and vascular homeostasis and modulates the phenotype of ECs [18]. Changes in the haemodynamic environment contribute to vasoregulation following acute changes in blood flow and vessel wall remodelling in response to chronic haemodynamic alteration [18]. It is well-established that alterations in the haemodynamic environment contribute to EC dysfunction, which is a pathological characteristic of the development of vascular diseases, including VGF [19]. EC communication is essential for maintaining the integrity of the endothelium and vascular homeostasis [20,21]. However, the effect of acute alteration in the haemodynamic environment following vein graft implantation on EC protein release and the resulting effects on ECs remain poorly characterised. Consequently, characterising the effect of altered FSS on EC protein release and its effect on ECs may help further define the molecular and mechanical basis of the development of VGF and provide pharmacological targets for the prevention of VGF.

## 2. Results

### 2.1. Exposure of vECs to Arterial FSS Induced a Pro-Inflammatory Phenotype and Increased Apoptosis

Confluent monolayers of HUVECs were pre-conditioned with low-magnitude FSS characteristic of the venous circulation (1.5 dynes/cm^2^) for 18 h to represent the vein in its native environment prior to vein graft implantation. HUVECs were subsequently exposed to venous or arterial (15 dynes/cm^2^) FSS for a further 24 h to represent the acute response of ECs following vein graft implantation into the arterial circulation. It is well-established that vECs exposed to arterial FSS demonstrate an upregulation of inflammatory markers [14,21]. Therefore, to validate that the vECs responded to the increase in FSS magnitude, RT-qPCR and Western blotting were used to quantify the expression of the inflammatory adhesion molecule intercellular adhesion molecule (ICAM)-1 and showed a significantly increased level of ICAM-1 mRNA (Figure 1A; *p* = 0.0312) and protein (Figure 1B; *p* = 0.0156) in HUVECs exposed to arterial FSS compared to HUVECs exposed to venous FSS. Furthermore, the nuclear factor kappa-light-chain-enhancer of activated B cells (NFκB) is a key mechanosensitive co-ordinator of the inflammatory response in ECs following exposure to an altered haemodynamic environment [22]. Therefore, NFκB (p65 subunit) was quantified to demonstrate induction of the pro-inflammatory phenotype. Significantly increased expression of the p65 subunit NFκB mRNA level following 2 h exposure to arterial FSS (Figure 1C; *p* = 0.0194) and NFκB p65 subunit protein following 24 h exposure to arterial FSS (Figure 1D; *p* = 0.0472) was detected compared to levels in HUVECs exposed to venous FSS.

EC injury and death following vein graft implantation is a pathological driver of dysfunction of the endothelium and initiator of VGF [15,23]. As a result, the effect of acute exposure of vECs to arterial FSS on the proportion of apoptotic cells was investigated using immunocytochemical analysis of CC3 as an indicator of apoptotic cells. Analysis of a confluent monolayer of HUVECs exposed to arterial-magnitude FSS for 24 h demonstrated a significant increase in the proportion of cells positive for CC3 in comparison to cells exposed to venous FSS (Figure 1E; *p* = 0.0026).

### 2.2. The Exposure of vECs to Arterial FSS Resulted in Increased Production and Release of FGL2

To determine the effect of applying arterial FSS to vECs on the EC secretome, conditioned media was collected from HUVECs pre-conditioned to venous FSS before further exposure to venous or arterial FSS for 24 h. The proteomic analysis of the conditioned media revealed 373 secreted proteins, 17 of which were significantly differentially secreted between the conditioned media derived from HUVECs exposed to arterial FSS or venous FSS (Appendix A). A literature search was carried out to determine the biological relevance of the 17 differentially secreted proteins to VGF, identifying fibroleukin (FGL2) as a protein of interest due to its emerging role as a contributor to EC dysfunction and cardiovascular disease (CVD) [24,25,26]. FGL2 was significantly increased by 1.30-fold in the secretome of HUVECs exposed to arterial FSS (*p* = 0.0493). This increase was validated by Western blot analysis of conditioned media derived from HUVECs exposed to arterial FSS, which showed a comparable increase in FGL2 compared to HUVECs exposed to venous FSS (Figure 2A; *p* = 0.0078).

Furthermore, to determine the effect of arterial FSS on FGL2 production, RT-qPCR was performed on lysates from pre-conditioned HUVECs exposed to arterial FSS for 24 h. RT-qPCR analysis showed that FGL2 transcription is also increased by the exposure of vECs to arterial FSS (Figure 2B; *p* = 0.0312).

### 2.3. Shear Stress-Induced Production and Release of FGL2 Was Mediated by NFκB Activity

Previous studies have linked the induction of FGL2 transcription to NFκB activity mediated by TNFα [27]. Therefore, HUVECs grown in static conditions were treated with recombinant TNFα (10 ng/mL) for 24 h, and FGL2 protein in the conditioned media was quantified. There was a significant increase in the release of FGL2 protein into conditioned media in HUVECs treated with TNFα compared to untreated controls (Figure 3A; *p* = 0.0203). Furthermore, to determine the contribution of NFκB activation to shear stress-induced FGL2 production and release, HUVECs pre-conditioned to venous FSS for 18 h were treated with the NFκB inhibitor BAY 11-7085 (BAY11, 1 µM) in the absence of flow for 45 min before exposure to arterial FSS for a further 24 h. Treatment with BAY11 resulted in a significant decrease in FGL2 mRNA levels (Figure 3B; *p* = 0.0444) and protein release into conditioned media compared to HUVECs treated with the vehicle control (Figure 3C; *p* = 0.0476).

### 2.4. FGL2 Induced Apoptosis in vECs Cultured in Static Conditions and vECs Exposed to Venous FSS

As previous studies demonstrated a role for FGL2 in apoptosis in a range of cell types [28], the role of FGL2 in FSS-induced vEC apoptosis was investigated by applying recombinant FGL2 (rFGL2, 20 ng/mL [29,30]; R&D Systems, 10303-FL-050) to HUVECs cultured in static conditions for 24 h. Immunocytochemical analysis of CC3 demonstrated a significant increase in the proportion of apoptotic HUVECs treated with rFGL2 in comparison to HUVECs (Figure 4A; *p* = 0.0140), comparable to apoptosis induced by recombinant TNFα (10 ng/mL; R&D systems, 210-TA) under the same conditions (n = 4; 10.91 ± 1.26% control cells vs. 15.72 ± 1.13% treated cells; *p* = 0.0317, two-tailed, paired t-test). Furthermore, Western blot analysis of cleaved poly (ADP-ribose) polymerase (PARP) showed a significant increase in cleaved PARP protein in static HUVECs treated with rFGL2 compared to untreated HUVECs (Figure 4B; *p* = 0.0312).

To determine the contribution of FGL2 to the increased levels of apoptosis observed in HUVECs exposed to arterial FSS, rFGL2 was added to HUVECs following the pre-conditioning step and was incubated with the ECs during the 24 h exposure to venous FSS. FGL2 protein significantly increased the percentage of apoptotic vECs, as determined using immunocytochemistry for CC3 (Figure 4C; *p* = 0.0133), in comparison to control vECs exposed to venous FSS.

### 2.5. FcγRIIB Was Detected in vECs Exposed to Venous and Arterial FSS

Previous studies have demonstrated that FGL2 binds to the receptor FcγRIIB to induce apoptosis in immune cells as part of its role as an immunomodulator [31]. To determine whether this receptor contributes to vEC apoptosis following exposure to arterial FSS, the presence of FcγRIIB in HUVECs exposed to venous or arterial FSS was investigated using RT-qPCR and Western blotting. It was observed that the receptor was present on HUVECs, but mRNA (Figure 5A; *p* = 0.3561) and protein levels (Figure 5B; *p* = 0.2946) were not modulated by the magnitude of FSS.

### 2.6. Blocking FcγRIIB Attenuated FGL2-Induced EC Apoptosis and ERK1/2 Signalling

Following the validation that FcγRIIB was present on HUVECs under flow, a FcγRII blocking antibody (2.5 µg/mL) was added to the media of HUVECs (cultured in static conditions) for 45 min before addition of rFGL2 (20 ng/mL) for 24 h and compared to control ECs treated with a species-matched IgG control due to the interaction between IgG binding the FcγRs (Appendix A).

The effect of blocking FcγRIIB on FGL2-induced apoptosis was investigated using CC3 immunocytochemistry and Western blotting for cleaved PARP. HUVECs pre-treated with the FcγRII blocking antibody prior to rFGL2 treatment demonstrated a significant decrease in CC3 in comparison to cells treated with the IgG control (Figure 6A; *p* = 0.0005). Likewise, Western blot analysis of cleaved PARP (a marker of apoptosis) showed a significant decrease in HUVECs pre-treated with the blocking antibody (Figure 6B; *p* = 0.0312).

Previous studies identified that overexpression of FGL2 is associated with increased phosphorylation of ERK1/2 (p-ERK1/2) and knockdown of FGL2 results in reduced p-ERK1/2 levels [27,32,33]. Consequently, the effect of rFGL2 addition to HUVECs (cultured in static conditions) on ERK1/2 activation was investigated using Western blotting. As expected, Western blot analysis identified that the application of rFGL2 resulted in a significant increase in p-ERK1/2 protein levels (Figure 6C; *p* = 0.0156), while there was no change in total ERK1/2, indicative of increased activation (Figure 6F). Subsequently, to determine whether the increase in ERK1/2 phosphorylation was associated with FGL2-FcγRIIB interaction, HUVECs cultured in static conditions were pre-incubated with the FcγRII blocking antibody for 45 min before treatment with rFGL2 for 24 h. Western blotting revealed there was a significant decrease in p-ERK1/2 protein levels in HUVECs pre-incubated with FcγRII blocking antibody, in comparison to HUVECs pre-treated with the IgG control (Figure 6D; *p* = 0.0096).

## 3. Discussion

Due to the role of ECs in detecting and communicating changes in the haemodynamic environment, ECs are dynamic and adaptable cells, which is reflected in their secretome with the release of factors allowing coordinated responses to mechanical stimuli [34]. It is well established that the communication between ECs is critical for the maintenance of vascular homeostasis and is altered in vascular disease [21]; however, the extent of dysfunctional intercellular signalling resulting from acute changes in the haemodynamic environment remains unclear. The increased release of FGL2 was identified from secretome analysis of vECs exposed to arterial FSS and was investigated due to growing evidence of its contribution to CVD [24,25,35].

FGL2 is a member of the fibrinogen-like protein family, which regulates a wide range of cell processes throughout the vasculature, including coagulation, cell adhesion, transendothelial migration, cell proliferation and regulation of transcriptions factors [27]. FGL2 occurs in a membrane-associated form, which has prothrombinase activity, and a secreted form, which acts as an immunomodulator to suppress T cell proliferation and inhibit dendritic cell maturation [27]. In addition, FGL2 protein has been linked to tumour progression by promoting proliferation and angiogenesis within the tumour [36]. The role of soluble FGL2 has been best characterised in regards to its immunomodulatory function, as it was first cloned from cytotoxic T lymphocytes and is thought to be primarily secreted by regulatory T cells [27]. Studies of ECs suggest that FGL2 primarily exists in the membrane-associated form in this cell type [25,37,38]. However, growing evidence suggests that secreted FGL2 may play a role in vascular pathology, a prominent example being hepatic ischaemia-reperfusion injury, wherein secreted FGL2 signalling promotes sinusoidal EC apoptosis and results in hepatocellular injury [28]. Secreted FGL2 is thought to primarily act through FcγRIIB for its immunomodulatory function to inhibit dendritic cell maturation, suppress T cell activation and induce apoptosis in B cells [27]. However, FcγRIIB is also expressed on ECs, and its activation has been previously shown to have adverse consequences, including antagonising nitric oxide production, leading to impairment of reparative mechanisms [39,40,41].

To date, the contribution of FSS in regulating the transcription and release of FGL2 has not been widely studied. Therefore, the focus of this study was to characterise the expression and release of FGL2 following the exposure of vECs to venous and arterial FSS and the related effects on vECs. Interestingly, previous studies have suggested that increased FGL2 transcription and circulating levels are associated with cardiovascular pathologies, including hypertension, whereby increased mechanical force on the vascular wall induces a remodelling response [42]. Furthermore, increased FGL2 expression has been identified in reperfusion injury following myocardial infarction, when severe stenosis contributing to ischaemia in the myocardium is treated with blood flow being restored following percutaneous coronary intervention (PCI) [43], and similarly in hepatic reperfusion injury during transplant, where blood flow is restored following organ implantation [28]. A common characteristic of these pathologies is the change in the haemodynamic environment from no FSS in transplanted vessels prior to implantation or reduced FSS in obstructed vessels to the sudden exposure to FSS following transplant implantation or PCI. As such, this pathological FSS profile is similarly observed in VGF, wherein ECs in the vein are chronically adapted to low-magnitude FSS before being exposed to a no-flow environment following vein dissection from the legs of patients undergoing CABG and the sudden exposure to high-magnitude FSS following vein graft implantation in the coronary artery circulation [19]. The presence of FGL2 in these vascular pathologies and the observation that the application of arterial FSS to vECs increased FGL2 production and release suggests an unexplored link between the regulation of FGL2 expression and release by FSS. Interestingly, the application of arterial FSS had a large effect on FGL2 mRNA production. As previously mentioned, FGL2 occurs as both a membrane-associated and a soluble protein [27]. Membrane-associated FGL2 is thought to contribute to thrombus development through the generation of thrombin resulting from its prothrombinase activity [25]. Early VGF is characterised by the occlusion of the vessel by thrombus formation in the first year post-surgery, in part due to the sudden exposure of the vein to arterial FSS [44]. As a result, the increase of FGL2 mRNA production following acute exposure to arterial FSS may also reflect changes in membrane-associated FGL2, which may be relevant to early VGF.

Indeed, it has previously been observed that TNFα-induced liver injury is dependent on FGL2 activity [28], and further studies have demonstrated the role of TNFα-mediated induction of FGL2 in a manner specific to ECs, whereas IFNγ stimulates FGL2 transcription in macrophages [31]. Moreover, TNFα stimulation of ECs cultured in static conditions resulted in a higher release of FGL2 into the conditioned media. TNFα acts as an activator and downstream effector of the NFκB signalling pathway, which has shown to be responsive to FSS and has a wide range of downstream effects, including the regulation of proliferation, differentiation, and immune response, which may link NFκB signalling to FGL2 transcription due to its role as an immunomodulator [27,39,45]. In this study, the addition of the NFκB inhibitor BAY11 reduced FGL2 production and release to the levels observed under venous FSS, with low levels of FGL2 having been previously demonstrated to be constitutively expressed [46], demonstrating a role for NFκB in the shear-dependent increase of FGL2 following exposure of vECs to arterial FSS.

Several studies have suggested that endothelial denudation/loss may result from the surgical trauma of LSV harvesting and surgical implantation during CABG surgery [15,23,47]. In addition, other ex vivo and in vitro studies have suggested that the onset of high-magnitude arterial FSS [48,49] and mechanical stretch can further exacerbate LSV injury by promoting EC apoptosis and reducing the protective properties of the endothelium [50,51]. Indeed, the acute exposure of vECs to arterial FSS increased the proportion of apoptotic ECs in comparison to ECs exposed to venous FSS. FGL2 has been reported to induce apoptosis in a range of cell types; therefore, the effect of FGL2 on endothelial apoptosis was investigated [31,52]. In line with other studies, treatment of HUVECs with rFGL2 at a comparable concentration observed in our FSS experiments demonstrated an increased proportion of apoptotic vECs. To further characterise this mechanism, the role of FcγRIIB in FGL2-induced endothelial apoptosis was investigated, as FGL2 is thought to interact with FcγRIIB and FcγRIII, with evidence supporting a role of FcγRIIB in FGL2-mediated apoptosis in immune cells [31,53]. Fcγ receptors (FcγRs) interact with the Fc region of IgG complexes to modulate intracellular signalling in immune cells [40]. Due to their interaction with the immune response, FcγRs regulate a range of inflammatory processes, resulting in a growing interest in their role in CVD [54]. FcγRIIB is the only known inhibitory member of the FcγR family, transmitting inhibitory signals through a cytoplasmic immunoreceptor tyrosine-based inhibitory motif (ITIM) [40]. As a result of its inhibitory role in controlling the strong inflammatory responses of immune cells and the resulting link to the inflammatory basis of CVD, FcγRIIB has been suggested to have a protective role in CVD in a number of preclinical animal models [55,56]. In contrast, evidence is increasingly demonstrating a pathological role of FcγRIIB signalling in CVD. Also, C-reactive protein (CRP) is associated with EC dysfunction and greater risk of CVD, with chronic elevations of CRP being associated with the development of hypertension [57]. Interestingly, studies have shown that CRP–FcγRIIB interaction attenuates the repair of the endothelium by antagonising endothelial nitric oxide synthase, which has resulted in increased interest in the targeting of FcγRIIB in CVD and inflammatory-related disease [58]. In this study, the presence of FcγRIIB in HUVECs was validated and showed that the receptor is present at detectable levels in HUVECs and FSS does not affect its level of expression. Although it must be considered that the blocking antibody was not specific to FcγRIIB, and therefore signalling through FcγRIIA may have also been suppressed, studies have only shown evidence of FGL2 interacting with FcγRIIB and FcγRIII [40,52,53]. Furthermore, the application of rFGL2 to HUVECs exposed to venous FSS resulted in an increase in the proportion of apoptotic cells, providing further evidence that FGL2 mediates endothelial apoptosis via FcγRIIB, which is present in ECs exposed to both venous and arterial FSS.

Evidence of ERK1/2 as a downstream effector of FGL2 signalling has been demonstrated in a range of studies, including in FGL2-induced ERK1/2-dependent autophagy in carcinoma cell lines [59], and the silencing of FGL2 as a target of interest in tumour development has been found to be associated with a decrease in ERK1/2 phosphorylation [60]. In addition, FGL2 overexpression has been linked with increased ERK1/2 phosphorylation [27,41], and the inhibition of ERK1/2 activity in a murine hepatitis model reduced the downstream functional activity of FGL2, with little effect on its production [54]. Studies have shown that aberrant activation of ERK1/2 can drive apoptosis through the cleaved caspase-3 pathway, although the mechanism by which this occurs remains unclear [61,62]. A range of MAPK/ERK kinase (MEK) inhibitors, which result in the inhibition of ERK1/2 phosphorylation, have been used to demonstrate a pro-apoptotic effect of sustained ERK1/2 signalling, as demonstrated by the presence of caspase-3 and PARP cleavage in a wide range of cell types [63]. In the physiological context, the activation of ERK1/2 is transient due to a feedback loop whereby ERK1/2 activates its own phosphatases, limiting its activity to minutes/hours [64]. In this study, elevated ERK1/2 phosphorylation was detected at 24 h, which suggests a mechanism of pro-apoptotic sustained ERK1/2 activation mediated by FGL2-induced activity. Interestingly, as an inhibitory FcγR, FcγRIIB signalling is associated with a decrease in ERK1/2 phosphorylation and signalling to reduce immune cell activation [65,66]. In contrast, in these experiments, blocking the FcγRIIB receptor resulted in reduced levels of phosphorylated ERK1/2 and apoptosis in ECs. However, there is evidence in other studies that interactions between FcγRIIB and other receptors can result in a change of function. For example, knockdown of FcγRIIB using silencing RNA in smooth muscle cells attenuated angiotensin-II-induced ERK1/2 phosphorylation due to its regulation of the internalisation of the Ang II type I receptor [41]. The effect of FGL2 on HUVEC apoptosis therefore may be mediated by the interaction of, or regulation by, FcγRIIB, with an additional receptor on the endothelium. Furthermore, in B cells, it has been demonstrated that homo-aggregation of FcγRIIB can induce a stress signal that is independent of the ITIM and leads to apoptosis [67].

In summary, the increased levels of apoptosis in vECs exposed to arterial FSS may therefore arise from the FSS-induced increased transcription, production and secretion of FGL2 protein and the subsequent binding to FcγRIIB, which may indirectly cause increased and sustained ERK1/2 activation, leading to EC apoptosis.

## 4. Materials and Methods

### 4.1. Culture of HUVECs

Primary human umbilical vein endothelial cells (HUVECs) pooled from male and female donors (Promocell, Heidelberg, Germany, C-12203) were cultured using Endothelial Cell Growth Medium (Promocell, C-22010) and penicillin and streptomycin (Gibco, Grand Island, NY, USA; 100 IU/mL and 100 mg/mL). Cells from 10 different batches of HUVECs were utilised in this study at passage 2–6, and n = 5–8 were used per experiment, as stated in figure legends.

### 4.2. Exposure of ECs to Fluid Shear Stress

HUVECs were seeded into the Ibidi µ-Slide I Luer 0.6 mm channel slide at a density of 1 × 10^6^ cells/mL and were allowed to adhere in static conditions for 2 h. Low-magnitude pre-conditioning FSS (1.5 dynes/cm^2^) was applied for 18 h using the Ibidi pump system, followed by exposure to the arterial- (15 dynes/cm^2^) or venous-magnitude (1.5 dynes/cm^2^) FSS for 24 h, using the PumpControl software (version 1.6.1) to generate the air pressure required to drive the flow of media for the given FSS value. For intervention experiments, HUVECs pre-conditioned to venous FSS for 18 h were treated with an NFκB inhibitor, BAY11 (Sigma-Aldrich, St. Louis, MO, USA, B5556; 1 µM), or a vehicle control while cultured under static conditions for 45 min before exposure to arterial FSS for a further 24 h.

### 4.3. Secretome Analysis

Conditioned media was collected from ECs pre-conditioned to venous FSS for 18 h and flowed under venous or arterial FSS for a further 24 h. The conditioned media was centrifuged (1000× *g*/10 min) to remove dead cells and debris and was subsequently concentrated ×40 using 10 K centrifugal concentrators (4000× *g*/30 min; Amicon, Merck, Darmstadt, Germany, UFC901024). Samples were incubated with an equal volume of 2,2,2-trifluoroethanol and heated at 60 °C for 1 h to support sample denaturation and the release of protein from vesicular structures. The samples were depleted of albumin and analysed using a Tandem Mass Tagging approach at the Proteomics Facility, University of Bristol. A Sequest search was performed against the Uniprot Human database, the Uniprot Bovine database and a ‘common contaminants’ database, filtered using a 5% false discovery rate cut-off. Bovine proteins and contaminants were excluded, and proteins of interest were highlighted by using a paired *t*-test to identify proteins with a significant log-fold change (*p* < 0.05).

### 4.4. Western Blotting

Cell lysates were collected using 1% (*w*/*v*) sodium dodecyl sulphate (SDS) lysis buffer, with protein content calculated from a standard curve using a BCA assay (Thermofisher Scientific, Waltham, MA, USA, 23235). Samples were treated with an equal volume of Laemmli buffer (Sigma-Aldrich, S3401) for analysis by SDS-PAGE and immunoblotting following a 5 min heat-denaturation incubation at 95 °C. For all Western blot experiments, the Biorad Mini format 1-D electrophoresis system (Bio-Rad, Hercules, CA, USA, 1658004) was used, with samples loaded into 4–15% Mini-PROTEAN^®^ TGX Stain-Free™ Protein Gels (Bio-Rad, 4568084) and exposed to electrophoresis at 200 volts for 30 min in 1 × Tris/Glycine/SDS (TGS) running buffer (Bio-Rad, 161-0772). To identify protein molecular weights, 2 µL of BLUeye Pre-Stained Protein Ladder (Geneflow, Lichfield, UK, S6-0024) was loaded alongside samples. Following electrophoresis, gels were imaged on the ChemiDoc XRS+ imaging system (Bio-Rad, 1708265), and ImageLab software (version 6.1.0) was used to carry out densitometric analysis of stain-free protein bands to act as a loading control. Following imaging of the gel, proteins were transferred using a Trans-Blot turbo nitrocellulose (0.2 µm) transfer pack (Bio-Rad, 170-4159) using the Trans-Blot turbo transfer system (Bio-Rad). To block non-specific binding, nitrocellulose membranes were incubated with 5% bovine serum albumin (BSA)/tris-buffered saline/tween (TBST) for 1 h. The membranes were incubated with the primary antibodies diluted in 5% (*w*/*v*) BSA/TBST for intercellular adhesion molecule-1 (ICAM-1, 1:300; Abcam, Cambridge, UK, 171123), NFκB (1:500; Cell Signalling Technology, Danvers, MA, USA, 8242), FGL2 (1:300; Abcam, ab198029), p-ERK1/2 (1:1000; Cell Signalling Technology, 9101), total ERK1/2 (1:500; Cell Signalling Technology, 4695), cleaved PARP (1:500; Abcam, ab32064), or FcγRII (1:500; Bio-Techne, Minneapolis, MN, USA, AF1330) at 4 °C overnight and were visualised with HRP-conjugated anti-IgG secondary antibodies diluted 1:2000 with 5% BSA/PBS. Enhanced chemiluminescent HRP-substrate (Merck, WBLUF0500) was used to detect bound antibodies. The chemiluminescence was visualised using the BioRad Chemidoc system, and protein levels were normalised to the stain-free gel for densiometric analysis performed with Image Lab Software (version 6.1.0, Bio-Rad).

### 4.5. RT-qPCR

Following exposure of HUVECs to FSS, RNA lysates were prepared using Qiazol lysis buffer (Qiagen, Hilden, Germany, 79306). RNA was extracted using a MiRNeasy Micro Kit (Qiagen, 217084) following the manufacturer’s instructions and converted to cDNA using a high-capacity RNA-cDNA kit (Applied Biosystems, Waltham, MA, USA, 4387406), according to the manufacturer’s instructions and using 200 ng RNA per sample. RT-qPCR was performed using a LightCycler 480 PCR machine using KiCq-start primers (Merck; Appendix A) with a SYBR green ready-to-use hot start reaction mix (Roche, Basel, Switzerland, 04707516001) for real-time PCR (Appendix A). A standard curve for the genes of interest ranging from 0.01 fg to 10 pg was prepared to calculate log(copy number/µg) or log(copy number/mg) for target gene quantification.

### 4.6. Immunocytochemistry

HUVECs cultured in static conditions and HUVECs exposed to FSS in the Ibidi channel slides were fixed using 3% (*w*/*v*) paraformaldehyde (PFA) for 15 min and permeabilised using 0.5% (*v*/*v*) Triton X-100. The samples were blocked using 5% BSA/phosphate buffered saline (PBS) for 1 h at room temperature, before incubation with anti-cleaved caspase-3 (CC3) antibody (1:100; Cell Signalling Technology, AF835) at 4 °C overnight. Following the incubation with the primary antibody, cells were washed with PBS and incubated with biotinylated anti-rabbit secondary antibody (Agilent Dako, Santa Clara, CA, USA, P021702-2) diluted 1:200 in PBS at room temperature (RT) for 1 h. Finally, cells were washed with PBS and incubated with the fluorescent label Streptavidin-488 (1:200; Vector Laboratories, Newark, CA, USA, SA-5488), and coverslips were mounted using Prolong Gold Antifade Reagent with 4′,6-diamidino-2-phenylindole (DAPI; Thermofisher, P36961) and left overnight at RT covered from light. CC3 and DAPI fluorescence was imaged using the EVOS FL Color Imaging System (Thermo Fisher Scientific, AMEFC4300) with a ×40 objective lens. Images were taken from randomly selected areas within 5 regions of the coverslip or channel slide, and total cell number was quantified using DAPI fluorescence on ImageJ (version V1.53u); cells positive for CC3 were counted, and data were expressed as percentage of CC3 positive cells.

### 4.7. Data Analysis

Data were expressed as mean ± standard error of the mean (SEM). Statistical analysis was performed using Graphpad Prism 9, and normality was assessed by using the Shapiro–Wilk and Kolmogorov–Smirnov normality tests. Where experiments consisted of only two groups, data were analysed using a paired, two-tailed *t*-test or Wilcoxon matched-pairs signed rank test. For experiments comparing more than two groups, a Friedman test with Dunn’s multiple comparison test (non-parametric data) was performed. The *p*-value was expressed where comparisons were made, with *p* < 0.05 considered significant. *p* * < 0.05, *p* ** < 0.01, *p* *** < 0.01.

## Figures and Tables

**Figure 1 ijms-25-07638-f001:**
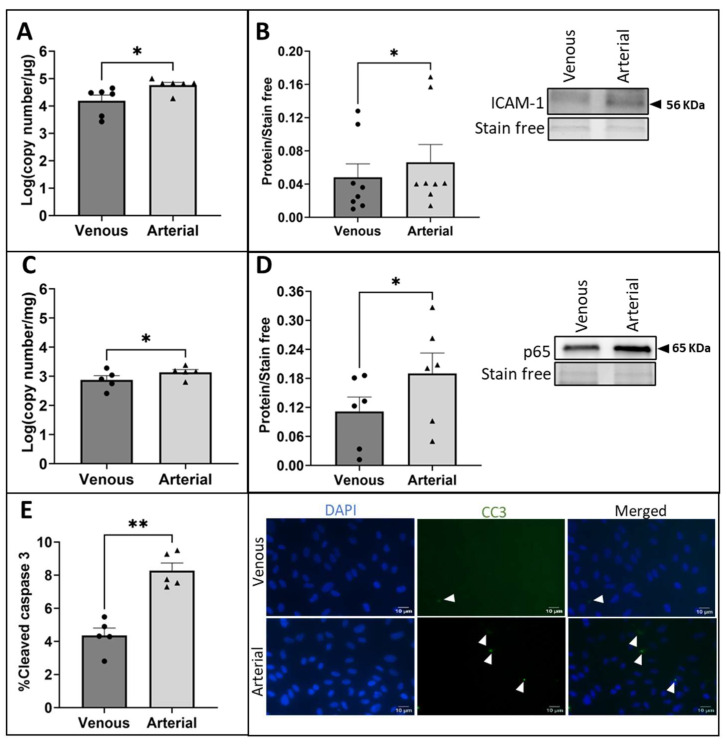
Pro-inflammatory activation and apoptosis of vECs exposed to arterial FSS. Confluent monolayers of HUVECs pre-conditioned to venous FSS for 18 h (1.5 dynes/cm^2^) were exposed to venous or arterial FSS (15 dynes/cm^2^) for 24 h in the Ibidi pump system. (**A**) ICAM-1 mRNA was quantified by RT-qPCR (n = 6; Wilcoxon matched-pairs signed rank test) and (**B**) ICAM-1 protein was quantified by Western blotting (representative Western blot and stain-free gel loading control; molecular weight indicated by black arrow; n = 8; Wilcoxon matched-pairs signed rank test). (**C**) NFκB p65 mRNA was quantified at 2 h by RT-qPCR (n = 5; two-tailed, paired *t*-test) and (**D**) NFκB p65 protein was quantified at 24 h (representative Western blot and stain-free gel loading control; molecular weight indicated by black arrow; n = 6; two-tailed, paired *t*-test). (**E**) Immunocytochemistry for CC3 (some positive cells (green) indicated by white arrowheads) was used to quantify the percentage of apoptotic ECs exposed to venous or arterial FSS for 24 h (nuclei stained blue with DAPI; n = 5; two-tailed, paired *t*-test). Values expressed as mean ± SEM. * *p* < 0.05, ** *p* < 0.01.

**Figure 2 ijms-25-07638-f002:**
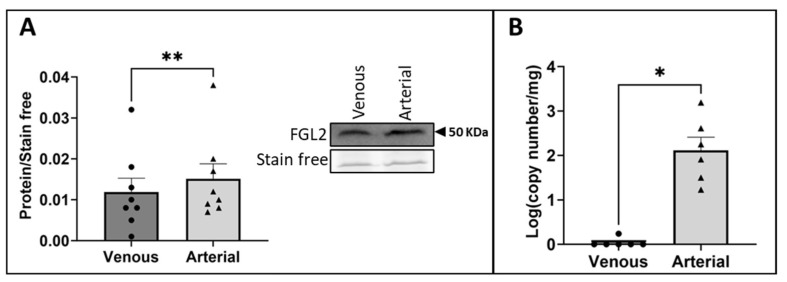
Shear stress-induced production and release of FGL2. HUVECs were pre-conditioned to venous FSS for 18 h before exposure to arterial or venous FSS for a further 24 h. (**A**) FGL2 protein in conditioned media was quantified by Western blotting (representative Western blot and stain-free gel loading control; molecular weight indicated by black arrow; n = 8; Wilcoxon matched-pairs signed rank test). (**B**) FGL2 mRNA level was analysed using RT-qPCR (n = 6; Wilcoxon matched-pairs signed rank test). Values expressed as mean ± SEM. * *p* < 0.05, ** *p* < 0.01.

**Figure 3 ijms-25-07638-f003:**
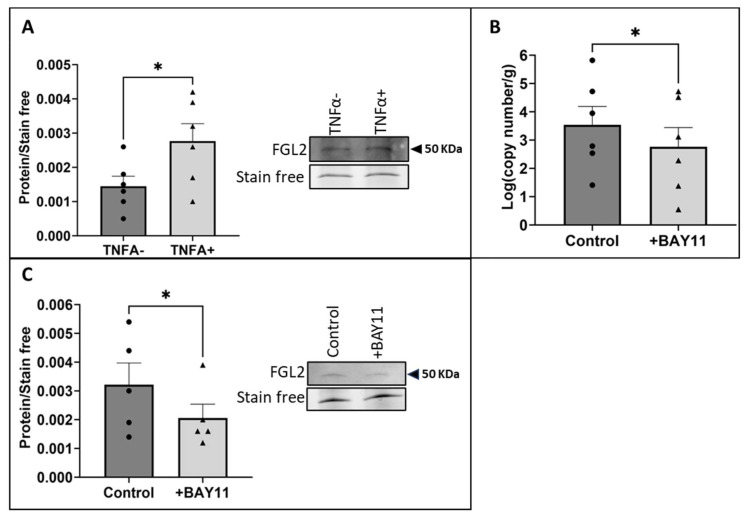
The role of NFκB in shear-induced FGL2 production and release. (**A**) Static HUVECs were treated with recombinant TNFα (10 ng/mL) for 24 h, and FGL2 protein release was quantified by Western blotting (representative Western blot and stain-free gel loading control; n = 6; two-tailed, paired *t*-test). HUVECs were treated with BAY11 (1 µM) or a vehicle control in static conditions for 45 min after venous pre-conditioning (18 h) and prior to exposure to arterial FSS for 24 h. (**B**) FGL2 mRNA was quantified by RT-qPCR (n = 6; two-tailed, paired *t*-test) and (**C**) FGL2 protein release was quantified by Western blotting of conditioned media (representative Western blot and stain-free gel loading control; n = 5; two-tailed, paired *t*-test). Values expressed as mean ± SEM. * *p* < 0.05.

**Figure 4 ijms-25-07638-f004:**
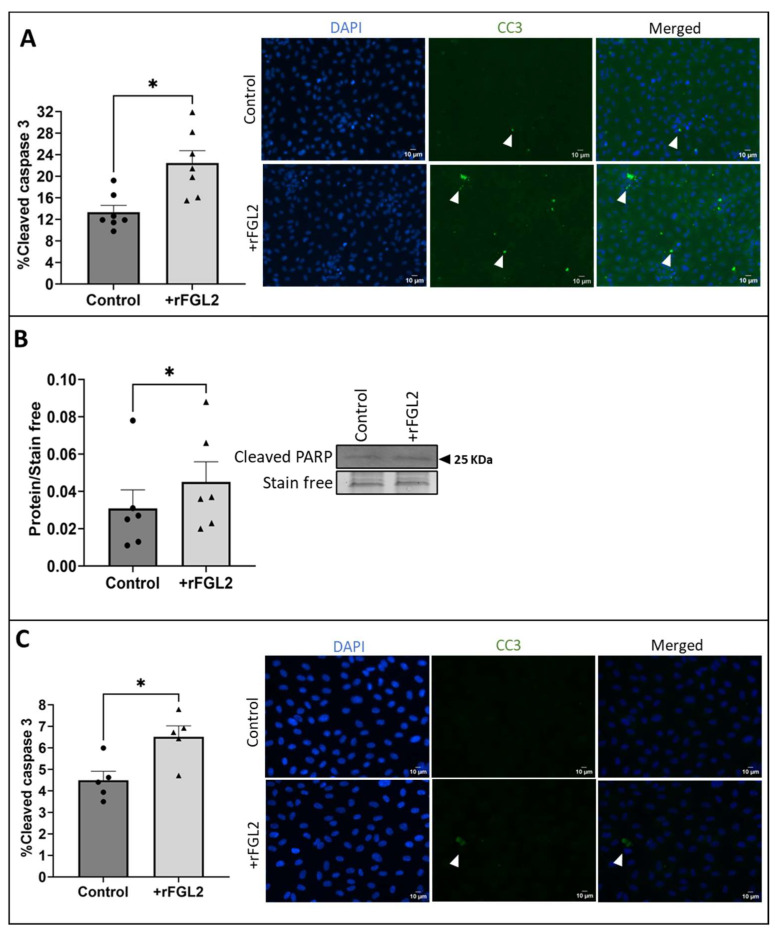
FGL2 production and release induced by arterial FSS and its role in endothelial apoptosis. (**A**) rFGL2 (20 ng/mL) was applied to static HUVECs for 24 h, and the percentage of apoptotic ECs was assessed by immunocytochemical analysis of CC3 (some positive cells indicated by white arrows; n = 7; two-tailed, paired *t*-test). (**B**) Apoptosis was quantified in HUVECs treated with rFGL2 by Western blotting for cleaved PARP (representative Western blot and stain-free gel loading control; molecular weight indicated by black arrow; n = 6; Wilcoxon matched-pairs signed rank test). (**C**) rFGL2 was added to HUVECs under venous FSS following exposure to pre-conditioning flow, and apoptosis was quantified by immunocytochemistry of the percentage of cells positive (green) for CC3 (n = 5; two-tailed, paired *t*-test). Nuclei stained blue with DAPI. Values expressed as mean ± SEM. * *p* < 0.05.

**Figure 5 ijms-25-07638-f005:**
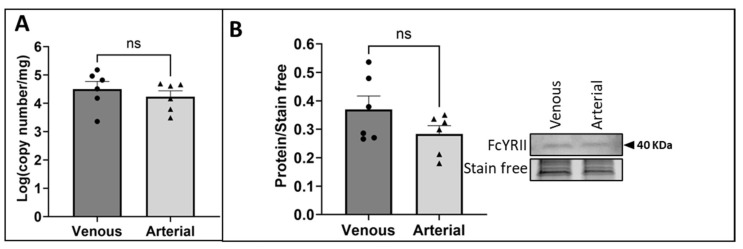
The presence of FcγRIIB in HUVECs exposed to FSS. HUVECs were exposed to venous or arterial FSS for 24 h. FcγRIIB mRNA and protein were quantified using (**A**) RT-qPCR and (**B**) Western blotting (representative Western blot and stain-free gel loading control; molecular weight indicated by black arrow; n = 6; two-tailed, paired *t*-test). Values expressed as mean ± SEM. ns indicates not significant.

**Figure 6 ijms-25-07638-f006:**
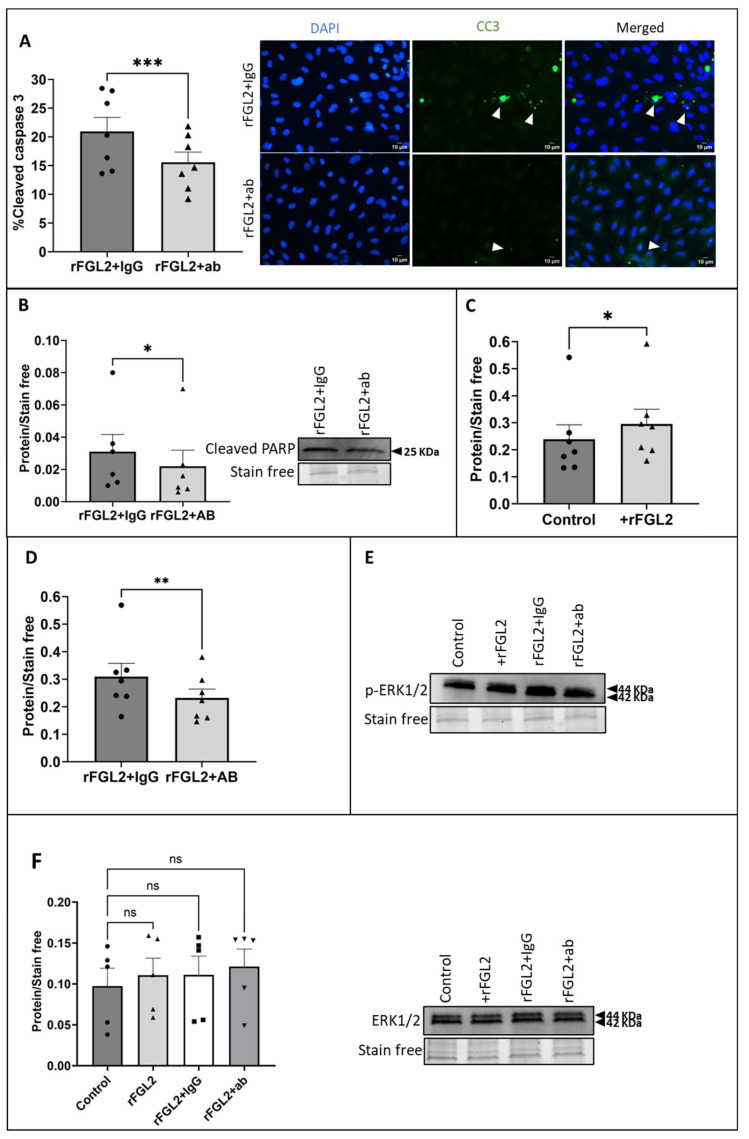
The role of FcγRIIB in FGL2-induced endothelial apoptosis. (**A**) A blocking antibody for FcγRII (2.5 µg/mL) was applied to HUVECs in static conditions for 45 min prior to treatment with rFGL2 (20 ng/mL) for 24 h, and the proportion of apoptotic ECs was quantified as the percentage of cells positive for CC3 (green; some indicated by white arrow; n = 7; two-tailed, paired *t*-test) and (**B**) Western blotting for cleaved PARP (representative Western blot and stain-free gel loading control; molecular weight indicated by black arrow; n = 6; Wilcoxon matched-pairs signed rank test). (**C**) Western blotting for p-ERK1/2 relative to stain-free gel loading control in HUVECs cultured in static conditions and treated with FGL2 or (**D**) FcYRII blocking antibody pre-treatment and FGL2. (**E**) Representative Western blot and stain-free gel loading control for panels C and D (n = 7; two-tailed, paired *t*-test). (**F**) The effect of treatment with rFGL2 and goat IgG or FcγRIIB blocking antibody on total ERK1/2 levels was quantified by Western blotting to validate that changes in ERK1/2 phosphorylation represent a method of determining the activation of ERK1/2 signalling (representative Western blot and stain-free gel loading control; molecular weight indicated by black arrow; n = 5; Friedman test with Dunn’s multiple-comparison test). Values expressed as mean ± SEM. ns indicates not significant. * *p* < 0.05, ** *p* < 0.01, *** *p* < 0.001.

## Data Availability

The mass spectrometry proteomics data have been deposited into the ProteomeXchange Consortium via the PRIDE [68] partner repository with the dataset identifier PXD050737 (reviewer account details: username: reviewer_pxd050737@ebi.ac.uk; password: HktZkKOe).

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
