# Peer review of "FGL2/FcγRIIB Signalling Mediates Arterial Shear Stress-Mediated Endothelial Cell Apoptosis: Implications for Coronary Artery Bypass Vein Graft Pathogenesis"

_ijms, 2024, doi:10.3390/ijms25147638_

Round 1

Reviewer 1 Report

Comments and Suggestions for Authors

This project examined how FGL2 and its putative receptor in endothelium mediate arterial fluid shear stress (FSS) induced apoptosis using an in vitro human endothelial cell model, and how this putative pathway underlies coronary artery bypass graft failure.  

Authors major observations were that HUVEC (EC) monolayers briefly preconditioned by venous FSS (18hrs) then changed to arterial aFSS for 24 h increased ICAM-1 expression and NFkB, as well as production/secretion of FGL2. Repeating same protocol and a pharmacological NFkB inhibitor prior to applying aFSS decreased secreted FGL2. Separately, addition of FGL2 to EC under static conditions or vFSS increased caspase 3 cleavage. Finally, FcgRIIB was implicated in this model using an antibody to this effect. Overall, the results are not convincing, as detailed below in comments, and the report lacks direct proof that FGL2 - FcgIIB is the mechanism involved in this model.

1.        Figs 1 and 5. Direct proof that FGL2 or FcgRIIB are responsible is lacking. Employing siRNA or crispr strategies to block expression of FGL2 and FcgRIIB  genes/proteins in HUVEC preconditioned and exposed to aFSS. Ab blocking or pharmacological approaches alone are not direct proof.  

2.        Fig 1E. The extend of activated caspase is not very impressive. How does this response compare to classic inducers of EC apoptosis-- stimuli like TNF-alpha or H2O2 etc? 4% vs 8% CC3 maybe statistically significant but is purely speculative without further in-depth studies.

3.        Fig 2. Data contradict one another. B panel qPCR data of FGL2 in venous FSS is minimal while FGL2 band in blot (A panel) is clearly detected (and not that different from arterial). This should be reconciled in results and discussion.  

4.        Fig 3. The amount of FGL2 released from EC as detected by WB in panel A is quite small. How does it compare to other constitutively secreted EC proteins? Bay11 (B panel) does not block FGL2 release, ~20% by my measurement. This finding is overinterpreted since the amount secreted and the reduction in FGL2 protein in conditioned media or message is so small.

5.        Fig 4. Amount of Apoptosis induced by FGL2 is not impressive.

6.        Methods. Suggest Authors repeat studies using low passage HUVEC, subculture 1 or 2, not beyond 3.

7.        Review references as there is duplicates, #29 same as #44.

Reviewer 2 Report

Comments and Suggestions for Authors

The authors describe the induction of FGL2 signaling in venous ECs upon arterial shear stress exposure, which might mediate cell apoptosis via caspase-3 signaling. Although the interest of the topic presents, the data presentation quality is quite low and extensive improvement must be conducted to support the conclusions. 

1. All the western blot results should show the housekeeping gene expression, such as GAPDH or beta-Actin. And the qRT-PCR results should be presented using deltaCT or delta-delta-CT methods that comparing to housekeeping genes.

2. Fig1D, for detecting p65 activation level, the author should also show p-p65 level and perform quantification by p-p65/total p65. Similarly Fig6E should also detect total ERK1/2 level.

3. Fig 2A, the secreted FGL2 in culture medium could be detected by ELISA for better quantification of the secretion level. The author should also clarify the application of the concentration of FGL2 (20 ng/mL) on HUVECs (Fig 4), whether such concentration is relevant to physiological or pathological conditions?

4. The percentage of CC3-positive cells in HUVECs is quite low. The author could consider use western blot to detect both the precursor caspase-3 and CC3 protein level. Apart from CC3 as the apoptosis indicator, other markers such as BrdU staining, p53/p21/BCL-2 expression, or cell viability by MTT assay should be included.

5. The rationel of investiagting FGL2 should be further addressed. The detection of FGL2 in the proteomic data should be labeled in Fig S1.

Comments on the Quality of English Language

NA

Round 2

Reviewer 2 Report

Comments and Suggestions for Authors

The manuscript has been improved and most questions are addressed.

Comments on the Quality of English Language

NA

Author Response

There are no Reviewers comments to respond to. We have however responded to the Academic Editor in the appropriate text box. Thank you.